# Heparan sulfate proteoglycans mediate prion-like α-synuclein toxicity in Parkinson's in vivo models

Merry Chen[1], Julie Vincent[1], Alexis Ezeanii[1], Saurabh Wakade[2], Shobha Yerigenahally[1] , Danielle E Mor[1]

**Parkinson's disease (PD) is a debilitating neurodegenerative disorder characterized by progressive motor decline and the aggregation of α-synuclein protein. Growing evidence suggests that α-synuclein aggregates may spread from neurons of the digestive tract to the central nervous system in a prion-like manner, yet the mechanisms of α-synuclein transmission and neurotoxicity remain poorly understood. Animal models that are amenable to high-throughput investigations are needed to facilitate the discovery of disease mechanisms. Here we describe the first *Caenorhabditis elegans* models in which feeding with α-synuclein preformed fibrils (PFFs) induces dopaminergic neurodegeneration, prion-like seeding of aggregation of human α-synuclein expressed in the host, and an associated motor decline. RNAi-mediated knockdown of the *C. elegans* syndecan *sdn-1*, or other enzymes involved in heparan sulfate proteoglycan synthesis, protected against PFF-induced α-synuclein aggregation, motor dysfunction, and dopamine neuron degeneration. This work offers new models by which to investigate gut-derived α-synuclein spreading and propagation of disease.**

## Introduction

Parkinson's disease (PD) is a debilitating movement disorder characterized by loss of dopaminergic neurons and formation of Lewy body inclusions containing aggregated α-synuclein protein (1, 2). Although the etiology of the disease remains unknown, growing evidence suggests that α-synuclein aggregation may originate in neurons that innervate the gastrointestinal tract, and spread to the central nervous system (CNS) via a prion-like mechanism (3, 4). Postmortem studies have found α-synuclein pathology in the enteric nervous system during early stages of PD (5), and gastrointestinal dysfunction is often among the first disease symptoms (6). Rodent models have recapitulated the spread of α-synuclein pathology with associated neurodegeneration and motor deficits after oral administration (7) or gastrointestinal injection (8, 9) of recombinant α-synuclein pre-formed fibrils (PFFs).

However, it remains unclear how α-synuclein originating in the digestive tract spreads to the CNS and induces neurodegeneration.

*Caenorhabditis elegans* offers a unique model system by which to address these questions, having orthologs for 60–80% of human genes, exceptionally high genetic tractability, and suitability to rapid, large-scale behavioral and phenotypic screening approaches (10, 11). The worm nervous system uses highly conserved signaling components to give rise to a complex set of behaviors (10). Similar to humans, *C. elegans* has an alimentary nervous system that facilitates feeding and acts largely independently from neurons in the rest of the body (termed the somatic nervous system) (12). Although some studies using *C. elegans* have shown α-synuclein cell-to-cell transmission within the somatic nervous system (13), or between neurons and other tissues (14, 15), it remains to be demonstrated whether α-synuclein of digestive origin can seed the aggregation of α-synuclein localized in body tissues and promote neurodegeneration in a prion-like manner.

Here, we have generated the first such models in *C. elegans*, by feeding worms human α-synuclein PFFs. We show that in animals expressing human α-synuclein ("host α-synuclein"), PFF feeding promotes α-synuclein aggregation in muscle cells and accelerates dopamine neuron degeneration. We find that several members of the heparan sulfate proteoglycan biosynthesis pathway mediate both PFF-induced α-synuclein aggregation and neurodegeneration, providing a novel platform by which to investigate prion-like activity of α-synuclein and neurotoxic mechanisms.

## Results

### Exogenous α-synuclein PFFs are ingested and spread to body tissues

Prions and prion-like proteins, including α-synuclein, exhibit self-templated replication or "seeding" via the repeated conversion of soluble protein into misfolded/aggregated conformations. This process requires endogenous protein as a source of aggregate amplification (16). Because *C. elegans* does not naturally possess an α-synuclein homolog, we took advantage of existing transgenic

[1]Department of Neuroscience and Regenerative Medicine, Medical College of Georgia at Augusta University, Augusta, GA, USA   [2]College of Sciences, Georgia Institute of Technology, Atlanta, GA, USA

Correspondence: dmor@augusta.edu

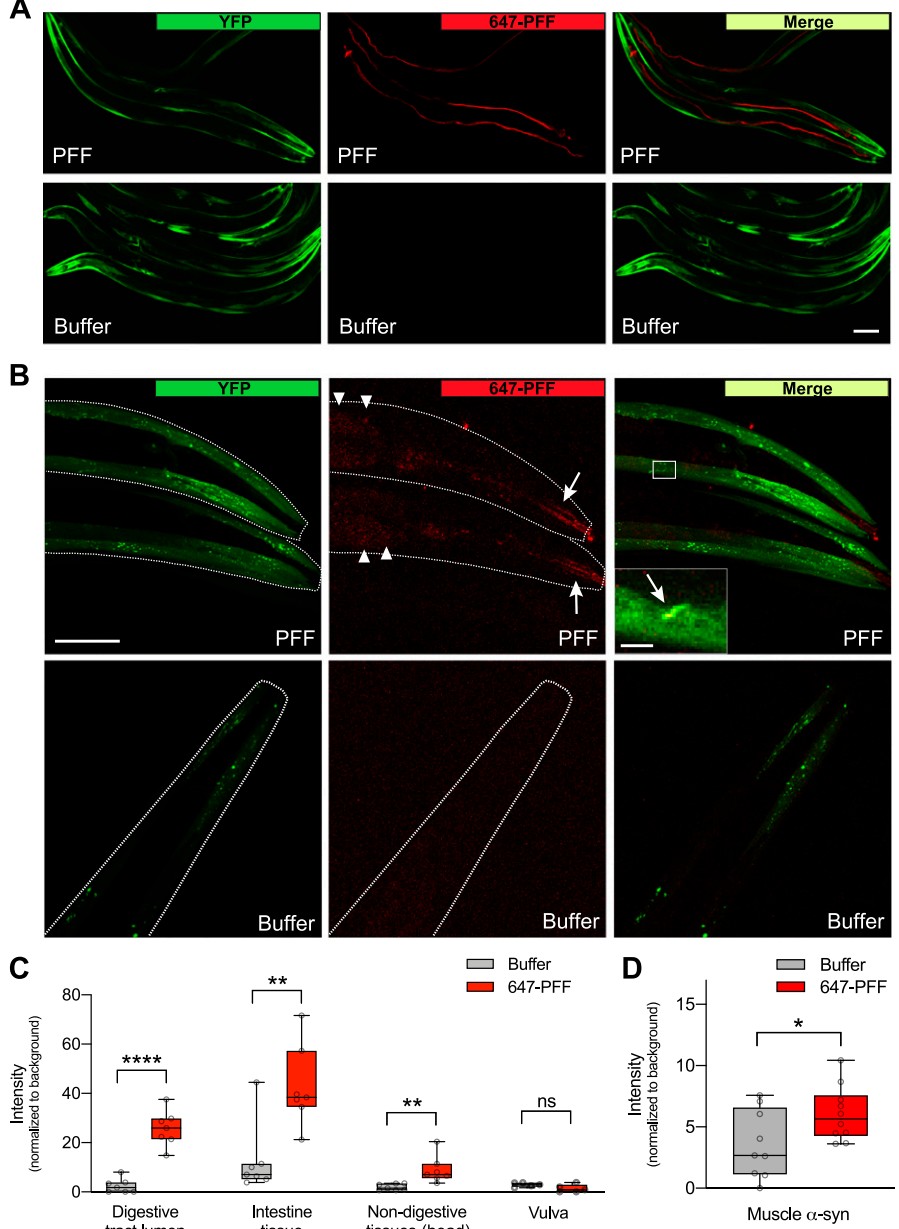

**Figure 1. α-synuclein pre-formed fibrils (PFFs) localize to the digestive tract and body tissues.**
**(A)** Worms expressing human wild-type α-synuclein–YFP in muscle cells (strain NL5901) were fed Alexa Fluor 647–labeled PFFs or buffer on day 1 of adulthood and imaged on day 2. Scale bar, 100 μm. **(B)** After a 1-h "washout" period to remove excess labeled PFFs from the gut, worms were imaged and 647-labeled PFFs were detected in the digestive tract lumen (arrows) and intestinal cells (arrowheads). Inset shows colocalization of host α-synuclein–YFP and 647-labeled PFFs. Scale bar, 50 μm; inset scale bar, 5 μm. **(C)** Quantification of Alexa Fluor 647 signal in multiple tissues showing that PFFs are present in digestive and other body tissues, but not vulva. n = 7 for all groups except n = 5 for vulva groups. Two-tailed *t* test. **(D)** Quantification of colocalization of Alexa Fluor 647 signal and α-synuclein–YFP in muscle cells. n = 9 for Buffer, 10 for PFF. Two-tailed *t* test. ns, not significant. *P < 0.05, **P < 0.01, ****P < 0.0001. Boxplots show minimum, 25th percentile, median, 75th percentile, and maximum.

strains expressing human wild-type α-synuclein in neurons (17) or muscle (18), for which there are only subtle degenerative phenotypes. This allowed us to ask whether α-synuclein PFF exposure through the digestive tract could accelerate disease phenotypes in neurons and other somatic tissues.

PFFs were generated from recombinant human wild-type α-synuclein using standard methods (19). Fibrillization was monitored by Thioflavin T and sedimentation analysis was used to confirm that the protein had become insoluble (Fig S1). To facilitate uptake of PFFs by ingestion, we pre-mixed freshly sonicated PFFs with the standard worm food source (*Escherichia coli*), and exposed animals to this mixture for 24 h starting on day 1 of adulthood. Adult-only PFF treatment was chosen to avoid any potential toxic effects on development, as well as to prevent PFFs from entering

through the worm cuticle, which becomes impermeable by day 1 of adulthood (20).

To confirm that exogenous PFFs are ingested, and investigate a possible spread to body tissues including potential localization with host α-synuclein, PFFs labeled with Alexa Fluor 647 dye were prepared and exposed to day 1 worms expressing human α-synuclein–YFP in muscle. On day 2 (after 24 h of exposure), PFFs were found localized to the digestive tract and the intestinal lumen, indicating that they are ingested (Figs 1A and S2A). After a 1-h "washout" period in which worms were allowed to clear the digestive tract of excess dye-labeled PFF, tissues were examined at higher magnification and the 647 Alexa Fluor signal was quantified (Fig 1B–D). Although some amount of PFFs were still present in the lumen of the digestive tract, 647-labeled PFFs were also detected in

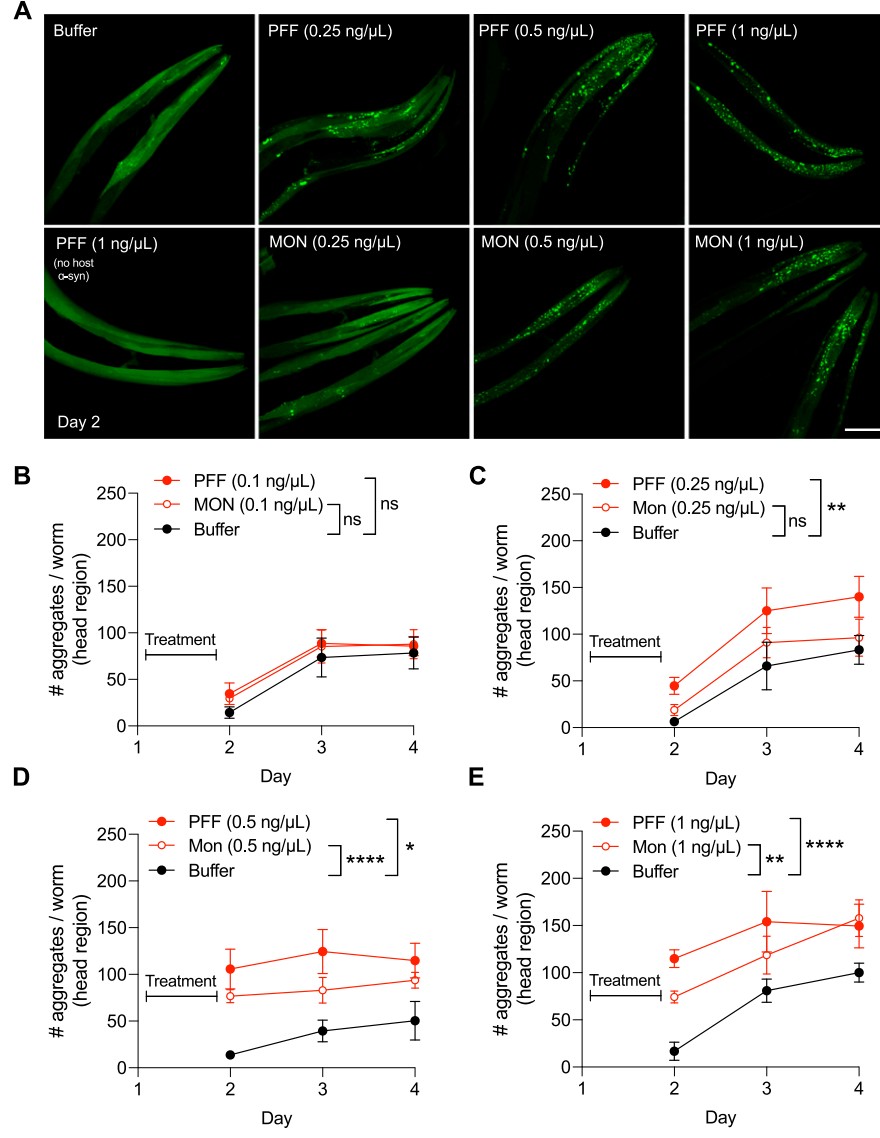

**Figure 2. Exogenous pre-formed fibrils (PFFs) promote the aggregation of host α-synuclein.**
**(A)** Worms expressing human wild-type α-synuclein–YFP in muscle cells (strain NL5901) were fed indicated dosages of PFFs or monomer (MON), or buffer alone, on day 1 of adulthood. Worms expressing YFP in muscle without transgenic α-synuclein expression (strain AM134) were also fed PFFs on day 1. On day 2, α-synuclein aggregates were imaged. Scale bar, 50 μm. **(B)** Quantification of # α-synuclein aggregates per worm in the head region after treatment with 0.1 ng/μl PFF or MON. n = 6 for each group on day 2 except n = 7 for MON, n = 8 for each group on day 3 except n = 10 for PFF, n = 7 for each group on day 4 except n = 9 for MON. Data are mean ± SEM. two-way ANOVA. **(C)** Quantification of # α-synuclein aggregates per worm in the head region after treatment with 0.25 ng/μl PFF or MON. For day 2, n = 6 for Buffer, 12 for PFF, 8 for MON. For day 3, n = 10 for Buffer, 9 for PFF, 13 for MON. For day 4, n = 7 for Buffer, 8 for PFF, 10 for MON. Data are mean ± SEM. Two-way ANOVA with Tukey's post hoc test. **(D)** Quantification of # α-synuclein aggregates per worm in the head region after treatment with 0.5 ng/μl PFF or MON. n = 7 for each group on day 2 except n = 6 for PFF, n = 9 for each group on day 3 except n = 8 for MON, n = 9 for each group on day 4 except n = 8 for MON. Data are mean ± SEM. Two-way ANOVA with Tukey's post hoc test. **(E)** Quantification of # α-synuclein aggregates per worm in the head region after treatment with 1 ng/μl PFF or MON. For day 2, n = 6 for Buffer, 9 for PFF, 10 for MON. For day 3, n = 9 for Buffer, 5 for PFF, 9 for MON. For day 4, n = 8 for Buffer, 9 for PFF, 8 for MON. Data are mean ± SEM. Two-way ANOVA with Tukey's post hoc test. ns, not significant. *$P <$ 0.05, **$P <$ 0.01, ****$P <$ 0.0001.

intestinal cells, and had infiltrated other non-digestive tissues in the head (Fig 1B and C). These findings are consistent with PFFs originating in the alimentary canal and spreading to other body tissues. It is unlikely that PFFs gained entry to the body through the vulva, as there was no 647 dye signal detected in the region of the vulva (Figs 1C and S2B).

Exogenous PFFs were also found to be colocalized with α-synuclein–YFP expressed in muscle (Fig 1B and D). These data suggest that PFFs are able to spread to muscle tissue, where they may interact with the host α-synuclein and promote aggregation in a prion-like manner.

## Gut-derived α-synuclein PFFs promote toxic aggregation of host α-synuclein

To test if PFFs of digestive origin are able to promote the aggregation of host α-synuclein in body tissues, we used the muscle-specific

α-synuclein–YFP strain to visualize and quantify α-synuclein aggregates. We tested a range of PFF dosages (0.1–1 ng/μl) and examined host α-synuclein aggregation at day 2 (immediately after 24 h of PFF exposure), as well as on day 3 and day 4 (Fig 2). On day 2, buffer-treated animals have very few aggregates, whereas treatment with PFFs caused a dose-dependent increase in the number of aggregates on this day (images in Fig 2A, quantification in Fig 2B–E). Treatment with the lowest dose tested (0.1 ng/μl PFF) did not increase aggregation above the baseline for this strain, which does show formation of aggregates over time as previously described (18) (Fig 2B). However, treatment with 0.25 ng/μl PFF showed a progressive increase in the number of aggregates from day 2 to day 4, significantly higher than baseline aggregation (Fig 2A and C). Importantly, feeding worms the equivalent dosage of α-synuclein monomer did not significantly increase host α-synuclein aggregation (Fig 2A and C), consistent with prion-like mechanisms (16), which require the initiating seed to be misfolded/aggregated.

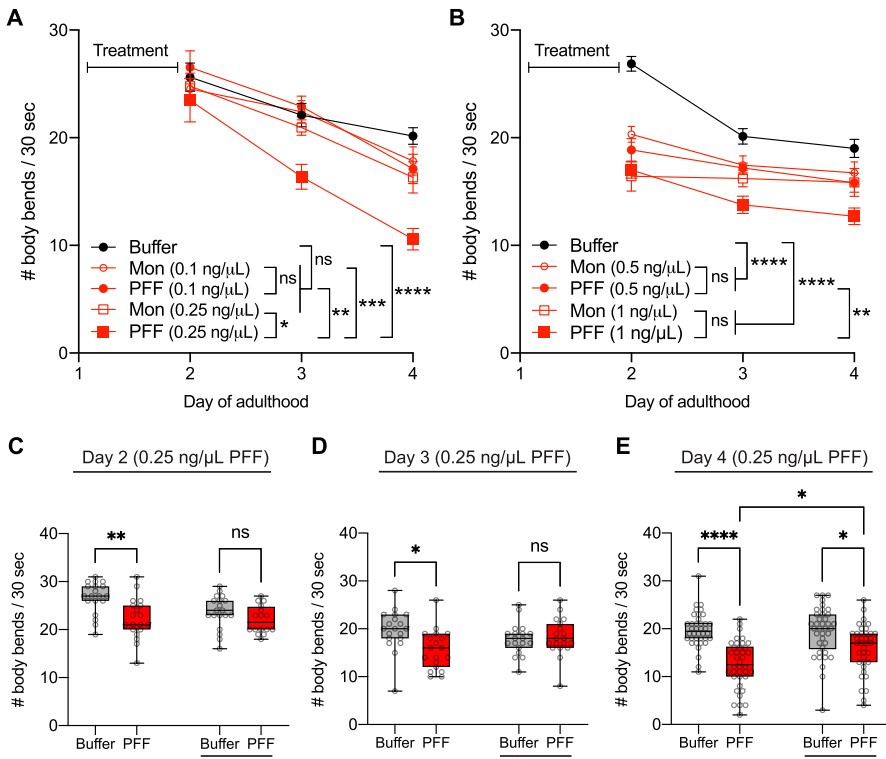

**Figure 3. Pre-formed fibril (PFF) feeding is associated with motor decline.**
**(A)** Worms expressing human wild-type α-synuclein–YFP in muscle cells (strain NL5901) were fed 0.1 or 0.25 ng/µl PFFs or monomer (MON), or buffer alone, on day 1 of adulthood and motor assays were conducted on days 2, 3, and 4. Data are mean ± SEM. n = 19 for each group. Two-way ANOVA with Tukey's post hoc test. **(B)** Worms expressing human wild-type α-synuclein–YFP in muscle cells (strain NL5901) were fed 0.5 or 1 ng/µl PFFs or monomer (MON), or buffer alone, on day 1 of adulthood and motor assays were conducted on days 2, 3, and 4. Data are mean ± SEM. n = 38 for each group. Two-way ANOVA with Tukey's post hoc test. **(C)** Motor assays on day 2 for α-synuclein–YFP (NL5901) and non-transgenic (N2) strains fed 0.25 ng/µl PFF or buffer. Data are mean ± SEM. n = 19 for each group except n = 16 for N2 PFF. Two-way ANOVA with Tukey's post hoc test. **(D)** Motor assays on day 3 for α-synuclein–YFP (NL5901) and non-transgenic (N2) strains fed 0.25 ng/µl PFF or buffer. Data are mean ± SEM. n = 19 for each group. Two-way ANOVA with Tukey's post hoc test. **(E)** Motor assays on day 4 for α-synuclein–YFP (NL5901) and non-transgenic (N2) strains fed 0.25 ng/µl PFF or buffer. Data are mean ± SEM. n = 38 for each group. Two-way ANOVA with Tukey's post hoc test. ns, not significant. *$P < 0.05$, **$P < 0.01$, ***$P < 0.001$, ****$P < 0.0001$.

In contrast to the 0.25 ng/µl dose of PFFs, the 0.5 and 1 ng/µl PFF treatments significantly increased aggregation as early as day 2, and did not show a progressive pattern over time (Fig 2A, D, and E). In both cases, the equivalent dosage of monomer also significantly increased aggregation (Fig 2A, D, and E), suggesting that these dosages exceed the threshold for producing prion-like effects. As an important control, we did not detect any aggregate-like puncta in worms expressing only YFP in muscle (no host α-synuclein) treated with the highest dosage of PFF, 1 ng/µl (Fig 2A), confirming that the puncta observed in all other cases are not an artifact of YFP expression.

Next, we asked whether increased aggregation in the muscle α-synuclein–YFP strain is associated with behavioral decline. To address this, we conducted crawling assays as a measure of basic motor function, and tested the same range of PFF concentrations (0.1–1 ng/µl) from day 2 to day 4. Consistent with a lack of aggregation-promoting ability of the lowest dose, 0.1 ng/µl PFF (Fig 2B), this dosage did not reduce motor function (Fig 3A). However, in line with our observations with aggregation, the 0.25 ng/µl concentration of PFFs caused a highly significant and progressive decline in motility compared with the baseline motor function of buffer-treated animals (Fig 3A). Moreover, worms fed 0.25 ng/µl PFF had significantly worse motor ability compared with those fed 0.25 ng/µl monomer, and this dosage of monomer did not exhibit a difference from buffer-treated controls (Fig 3A). This suggests that the progressive loss of motor function induced by 0.25 ng/µl PFF is prion-like, and is associated with the similar pattern of aggregation pathology observed in these animals.

The movement ability of worms fed 0.5 and 1 ng/µl PFF (Fig 3B) also paralleled the observations for these dosages with aggregation (Fig 2D and E). As early as day 2, both 0.5 and 1 ng/µl PFF-fed worms showed a drop in motor functioning, which remained relatively stable over the time-period tested (Fig 3B). Although these PFF concentrations produced highly significant motor decline compared with buffer-treated animals, the equivalent dosages of monomer also produced a significant decline, and there was no difference between PFF and monomer at each of these dosages (Fig 3B). Therefore, consistent with the non-prion–like induction of host α-synuclein aggregation with the 0.5 and 1 ng/µl exposures, these concentrations appear to also produce non-prion–like reductions in movement ability.

Given these data, indicating that the 0.25 ng/µl PFF dose may be optimal for modeling prion-like mechanisms in this *C. elegans* strain, we moved forward with this dose. We wished to further examine whether 0.25 ng/µl PFF causes motor decline in non-transgenic animals that have no host α-synuclein. On day 2, whereas PFF-fed worms expressing host α-synuclein in muscle showed a significant 17% reduction in average motility compared with buffer-treated controls, non-transgenic worms fed PFFs showed no deficit at this time-point (Fig 3C). On day 3, there was a significant 21% decline in average motility in α-synuclein–expressing animals fed PFFs, with no effect in non-transgenic animals (Fig 3D). Only by day 4 did we detect a reduction of motor function in PFF-fed non-transgenic worms, specifically a 17% loss of average motility (Fig 3E). At this time-point, however, PFF-fed worms expressing host α-synuclein had a 36% loss of motor function, and this group was found to have significantly worse

motility than PFF-fed non-transgenic worms (Fig 3E). Therefore, PFF toxicity is enhanced in animals expressing host α-synuclein, consistent with prion-like underlying mechanisms.

## α-Synuclein PFF feeding induces dopaminergic neurodegeneration

To test if PFF feeding can promote neurodegeneration of dopaminergic cells, we used a strain of worms expressing pan-neuronal human wild-type α-synuclein, which on its own has extremely subtle degenerative phenotypes (17). *C. elegans* possess six dopaminergic neurons in the head, four of which are of the CEP class, and two are of the ADE class. To quantify neurodegeneration, the cell body area of each dopaminergic cell type was measured, and the number of cell bodies of wild-type (normal) size was counted per worm. In addition, dopaminergic neurites were scored for normal, fragmented, or blebbed morphologies. We began by testing the 0.25 ng/μl PFF dosage because this was the optimal concentration to produce prion-like effects in the muscle-specific α-synuclein strain. However, treatment with 0.25 ng/μl PFF produced only mild effects in the pan-neuronal α-synuclein strain, even as late as day 5 (Fig S3A–D). Specifically, the average CEP cell body area of PFF-fed worms was reduced by 15% compared with buffer-treated controls, and there was no reduction in ADE cell body size (Fig S3B and C). While there was a significant loss of CEP cell bodies per worm, the degeneration observed in CEPs was also caused by 0.25 ng/μl monomer (Fig S3B). Despite the lack of degenerative changes in non-transgenic worms fed 0.25 ng/μl PFF (Fig S3E–G), we decided not to move forward with this dosage in the pan-neuronal α-synuclein strain.

Increasing the PFF dosage to 0.5 ng/μl produced more severe effects and allowed us to identify an optimal set of conditions for prion-like toxic phenotypes in this strain. As early as day 2, 0.5 ng/μl PFF treatment caused a 26% reduction in the mean CEP cell body area, and a corresponding significant loss of CEP cell bodies (Fig 4A and B). There was also a 52% drop in ADE cell body size in PFF-fed animals compared with buffer-treated controls, and a corresponding significant loss of ADE cell bodies (Fig 4A and C). Exposure to the equivalent dose of monomer did not produce significant degenerative effects in either CEPs or ADEs (Fig 4A–C), consistent with prion-like mechanisms driving neuronal injury at the day 2 time-point. Dopaminergic neurites, however, were spared on day 2 (Fig 4D).

By day 5, 0.5 ng/μl monomer showed similar toxicity to PFF (Fig 4E and F). Specifically, there was a 32% and 28% reduction in the mean CEP cell body area in PFF- and monomer-fed animals, respectively, and there was a significant loss of CEP cell bodies in both treatment groups (Fig 4F). There was no effect on ADE cell body area or number of cells (Fig 4G), potentially because of degeneration occurring by this time-point from α-synuclein expression in the background. Interestingly, neurites showed significantly more degenerated morphologies with PFF feeding than with monomer (Fig 4H). However, because of the monomer toxicity in CEP cell bodies, it appears that day 5 is too late to produce prion-like toxic effects at this dosage. Instead, the earlier time-point of day 2 recapitulated toxicity in both CEP and ADE cell types with PFF treatment and not monomer, suggesting that these are the optimal conditions to

model prion-like disease phenotypes in this strain. Importantly, the 0.5 ng/μl PFF dosage was insufficient to produce neurodegeneration in non-transgenic worms, even as late as day 5 (Fig S4).

## Heparan sulfate proteoglycans mediate PFF-induced toxicity

Although the precise mechanisms by which gut-derived α-synuclein PFFs enter other tissues are not known, evidence from mammalian cell culture studies suggests that PFFs can interact with heparan sulfate proteoglycans (HSPGs) on the cell surface and become endocytosed (21). However, HSPGs have yet to be tested as potential mediators of α-synuclein neurotoxic spread in vivo. To test if HSPGs may play a role in our new PFF-based *C. elegans* models of PD, we performed a targeted screen of seven HSPG pathway genes. Worms expressing human α-synuclein–YFP in muscle were fed RNAi corresponding to each HSPG gene starting at the L4 stage, to achieve adult-only gene knockdown. On day 1 of adulthood, animals were fed 0.25 ng/μl PFFs for 24 h, and then motor function was assessed on day 4 since this PFF dosage over this time period results in severe, prion-like motor decline in this strain (Fig 3A). As expected, PFF treatment on empty vector RNAi (Vect) significantly reduced motility compared with Vect RNAi animals treated with buffer (Fig 5A).

Strikingly, five of seven HSPG genes were found to be required for PFF-induced motor dysfunction (Fig 5A). In addition to *SDC1/sdn-1*, the cell surface membrane-bound proteoglycan syndecan, the HSPG biosynthetic enzyme genes *EXT1/rib-1* (exostosin glycosyltransferase 1), *EXTL3/rib-2* (exostosin like glycosyltransferase 3), *NDST1/hst-1* (N-deacetylase and N-sulfotransferase 1), and *HS3ST6/hst-3.2* (heparan sulfate-glucosamine 3-sulfotransferase 6) all significantly rescued motor function when knocked down by RNAi (Fig 5A). The two genes that had no effect, *GLCE/hse-5* (glucuronic acid epimerase), and *HS2ST1/hst-2* (heparan sulfate 2-O-sulfotransferase 1) were verified to have been knocked down successfully (Fig S5A). It was also determined that knockdown of each of the five positive hits in buffer-treated animals did not improve motor function (Fig S5B), indicating that the protective effects are PFF-dependent.

To further test if these genes regulate susceptibility to PFF-induced aggregation of host α-synuclein, day 4 animals raised on RNAi and fed 0.25 ng/μl PFFs were imaged and α-synuclein aggregates in the muscle were quantified. Three of these five genes – *sdn-1*, *rib-1*, and *hst-1* – when knocked down, achieved a remarkable 73–83% reduction in the number of α-synuclein aggregates compared with PFF-fed animals on Vect (Fig 5B and C). Importantly, these effects were PFF-dependent, in that the reductions seen in PFF-fed RNAi-treated animals were significantly larger than those of buffer-exposed RNAi-treated animals that were compared with buffer-exposed Vect (Fig 5C).

Finally, to determine if the three HSPG genes that regulate PFF-induced toxicity and seeding of aggregation in the muscle also play a role in PFF acceleration of dopaminergic neuron degeneration, worms expressing neuronal α-synuclein were crossed to a neuronal RNAi-sensitive background and raised on HSPG gene RNAi's starting at L4. On day 1, the worms were treated with 0.5 ng/μl PFFs, and dopaminergic neurons were assessed for neurodegeneration

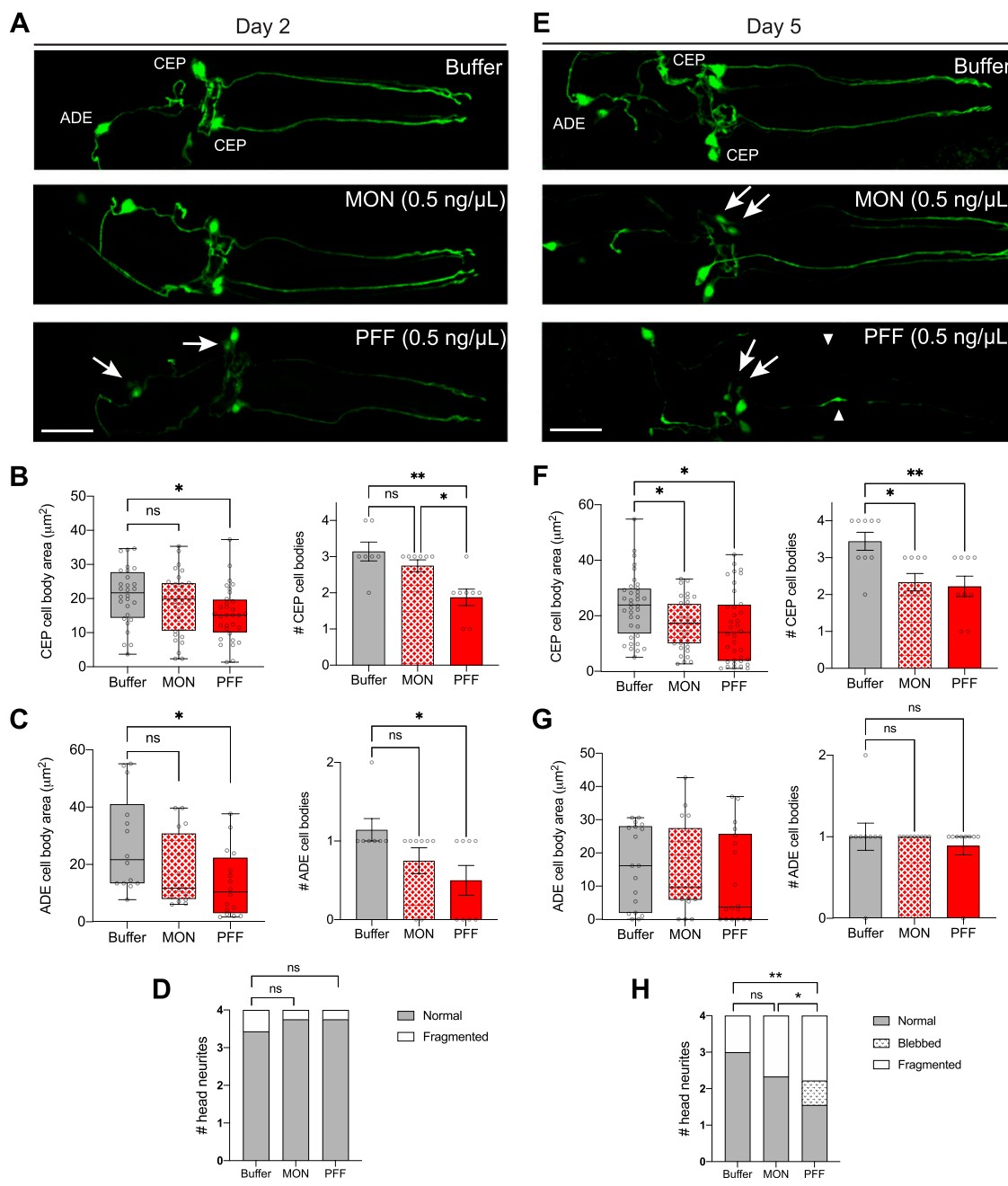

**Figure 4. α-synuclein pre-formed fibrils (PFFs) accelerate dopaminergic neurodegeneration.**
**(A)** Worms expressing pan-neuronal human wild-type α-synuclein (strain UM0011) were treated with 0.5 ng/μl PFFs or monomer (MON), or buffer alone, on day 1 of adulthood. On day 2, dopamine neurons were imaged and scored for degenerating cell bodies (arrows) and neurites. Scale bar, 20 μm. **(B)** Quantification of CEP cell body area, and number of CEP cell bodies on day 2 after 0.5 ng/μl treatment. Data for number of cell bodies are mean ± SEM. For cell body area, n = 32 for each group except n = 28 for Buffer. For number of cell bodies, n = 8 for each group except n = 7 for Buffer. One-way ANOVA with Tukey's post hoc test. **(C)** Quantification of ADE cell body area, and number of ADE cell bodies on day 2 after 0.5 ng/μl treatment. Data for number of cell bodies are mean ± SEM. For cell body area, n = 16 for each group except n = 14 for Buffer. For number of cell bodies, n = 8 for each group except n = 7 for Buffer. One-way ANOVA with Tukey's post hoc test. **(D)** Quantification of neurites on day 2 after 0.5 ng/μl treatment. n = 32 for each group except n = 28 for Buffer. Chi-squared test. **(E)** Worms expressing pan-neuronal human wild-type α-synuclein (strain UM0011) were treated with 0.5 ng/μl PFFs or monomer (MON), or buffer alone, on day 1 of adulthood. On day 5, dopamine neurons were imaged and scored for degenerating cell bodies (arrows) and neurites (arrowheads). Scale bar, 20 μm. **(F)** Quantification of CEP cell body area, and number of CEP cell bodies on day 5 after 0.5 ng/μl treatment. Data for number of cell bodies are mean ± SEM. For cell body area, n = 36 for each group. For number of cell bodies, n = 9 for each group. One-way ANOVA with Tukey's post hoc test. **(G)** Quantification of ADE cell body area, and number of ADE cell bodies on day 5 after 0.5 ng/μl treatment. Data for number of cell bodies are mean ± SEM. For cell body area, n = 18 for each group. For number of cell bodies, n = 9 for each group. One-way ANOVA with Tukey's post hoc test. **(H)** Quantification of neurites on day 5 after 0.5 ng/μl treatment. n = 36 for each group. Chi-squared test. ns, not significant. *P < 0.05, **P < 0.01. Boxplots show minimum, 25th percentile, median, 75th percentile, and maximum.

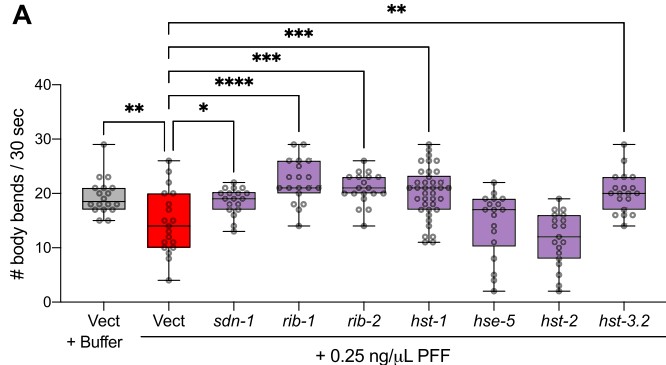

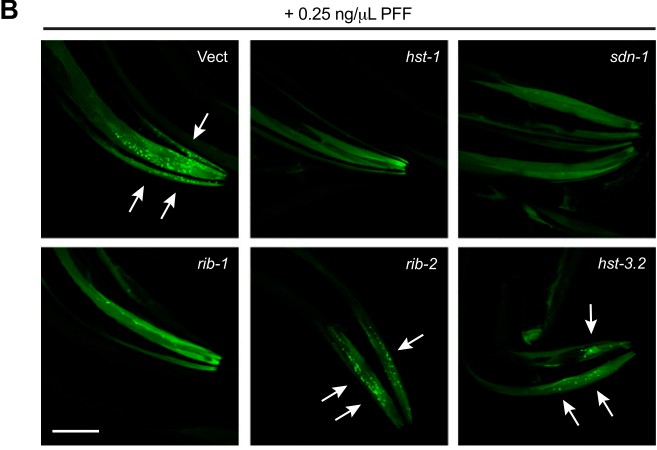

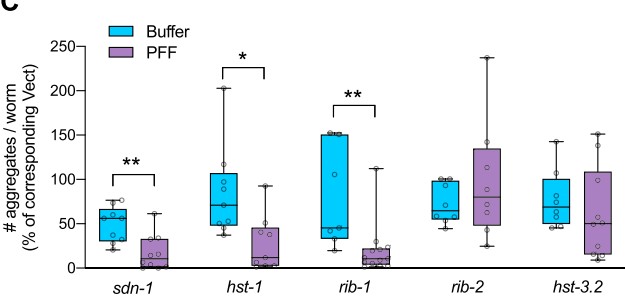

**Figure 5. HSPG gene knockdown rescues motor function and reduces α-synuclein aggregation in pre-formed fibril (PFF)-fed animals.**
**(A)** Worms expressing human wild-type α-synuclein–YFP in muscle cells (strain NL5901) were fed 0.25 ng/µl PFFs or buffer on day 1 of adulthood and motor assays were conducted on day 4. n = 38 for *hst-1* and *19* for each of the other groups. One-way ANOVA with Dunnett's post hoc test. **(B)** On day 4, α-synuclein aggregates (arrows) were imaged. Scale bar, 50 *µm*. **(C)** Quantification of # α-synuclein aggregates per worm normalized to the corresponding Vect group: values for PFF-fed groups are expressed as % of PFF-fed Vect control group, and values for Buffer-treated groups are expressed as % of Buffer-treated Vect control group. n = 9 each for *sdn-1* Buffer and *hst-1* groups, n = 8 each for *hst-3.2* Buffer and *rib-2* groups, n = 7 for *rib-1* Buffer, n = 10 each for *sdn-1* PFF and *hst-3.2* PFF, n = 13 for *rib-1* PFF. Two-tailed *t* test. Vect, empty vector RNAi. *$P < 0.05$, **$P < 0.01$, ***$P < 0.001$, ****$P < 0.0001$. Boxplots show minimum, 25th percentile, median, 75th percentile, and maximum.

on day 2. This PFF dosage and timing corresponds to the optimal conditions in which prion-like effects can be documented in the strain expressing neuronal α-synuclein (Fig 4A–D). We found that all three HSPG genes that afforded protection from prion-like PFF

phenotypes in worms expressing muscle α-synuclein significantly increased CEP cell body area in PFF-fed neuronal α-synuclein–expressing worms (Fig 6A and B). Again, this effect could not be accounted for by potential neuroprotection of the RNAi in buffer-treated animals (Fig 6B). The number of CEP cell bodies was also significantly higher in PFF-fed animals with *sdn-1*, *rib-1*, or *hst-1* knockdown, compared with PFF-fed animals on Vect (Fig 6C).

In terms of ADE cell body area, only *rib-1* afforded significant protection when compared with the effects of the RNAi in buffer-treated worms (Fig 6D), yet this may be due to the large number of ADEs that were not present (scored as having zero area) in all groups. Indeed, when we quantified the number of ADE cell bodies per worm, we found that knockdown of each of the three HSPG genes significantly increased the number of ADEs in PFF-fed animals (Fig 6E). Importantly, the effects of the RNAi on buffer-treated animals showed no differences for either CEP or ADE cell number when compared with buffer treatment on Vect (Fig S6). Collectively, these results suggest that *sdn-1*, *rib-1*, and *hst-1* play important roles in vivo in prion-like toxicity of α-synuclein derived from the digestive tract.

## Discussion

Here we offer new *C. elegans* models of PD based on gut-derived α-synuclein spreading, with resulting dopamine neuron degeneration and toxic seeding of host α-synuclein aggregation (Fig 7A). To our knowledge, this is the first in vivo demonstration that prion-like α-synuclein toxicity depends on the HSPG pathway, specifically *SDC1/sdn-1*, *EXT1/rib-1*, and *NDST1/hst-1* (Fig 7B). In PD patients, the progressive buildup of α-synuclein pathology is consistent with prion-like α-synuclein transmission (3), and the observation of α-synuclein positive inclusions in the enteric nervous system is suggestive of a "gut-to-brain" disease progression (5). Moreover, one of the earliest regions in the CNS to develop α-synuclein pathology is the dorsal motor nucleus of the vagus, which connects the brain to the periphery (4). Yet, despite growing evidence for prion-like α-synuclein spreading and toxicity in PD, the mechanisms of this potential spread and observed neuronal cell death remain unknown.

Several lines of evidence from animal models support the notion that α-synuclein cell-to-cell transmission may be a critical factor in neurodegenerative disease. In rodents, intracerebral inoculation of either α-synuclein PFFs (22), brain tissue from symptomatic A53T mutant α-synuclein transgenic mice (23), or brain tissue from synucleinopathy patients (24) results in widespread deposition of Lewy body–like α-synuclein inclusions often accompanied by neurodegeneration and motor deficits. Several studies have also demonstrated the spread of α-synuclein pathology from peripheral sites to the CNS, using intravenous (25), intramuscular (26), or intraperitoneal (27) injections of recombinant α-synuclein aggregates into rodents. More recently, oral administration or gastrointestinal injection of PFFs in mice was shown to recapitulate gut-to-brain α-synuclein spread and PD-like disease (7, 8, 9).

In *C. elegans*, a small number of studies have shown cell-to-cell transmission of α-synuclein. Using a split-GFP approach in distinct neuronal subsets, Tyson et al showed that α-synuclein could be transferred from neuron-to-neuron (13). A similar approach

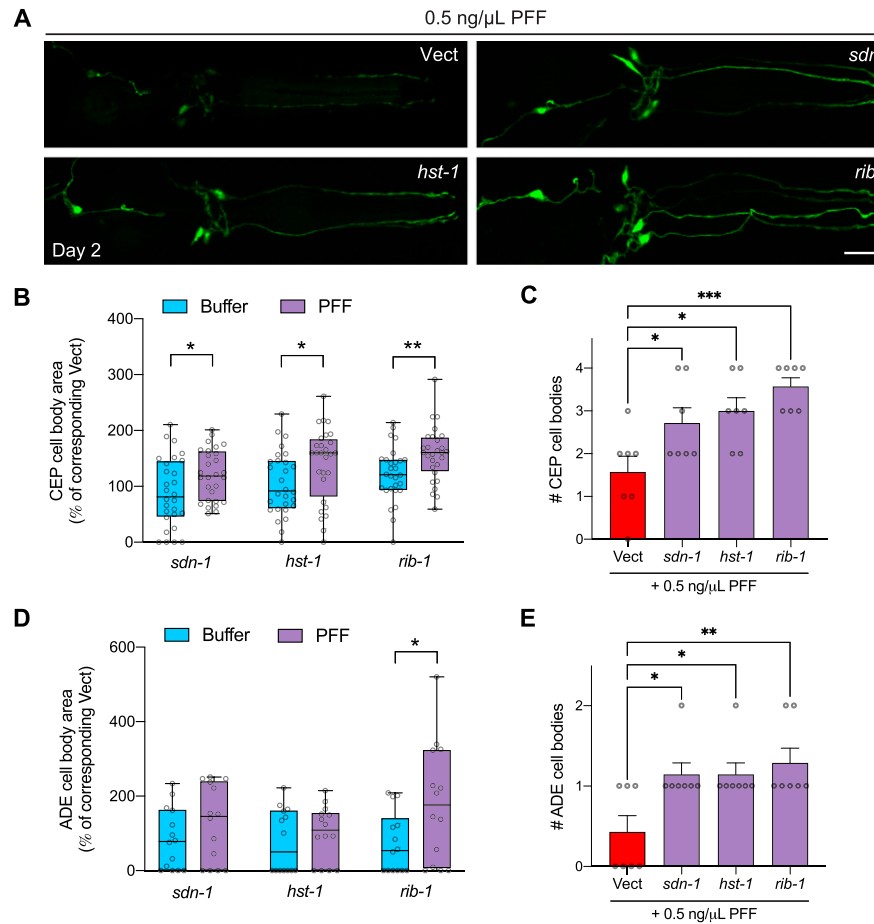

**Figure 6. Pre-formed fibril (PFF)–induced dopaminergic neurodegeneration is mitigated by HSPG gene knockdown.**
**(A)** Worms expressing pan-neuronal human wild-type α-synuclein in a neuronal RNAi-sensitive background (strain MOR002) were treated with 0.5 ng/μl PFFs on day 1 of adulthood. On day 2, dopamine neurons were imaged and scored for rescue of cell bodies. Scale bar, 20 μm. **(B)** Quantification of CEP cell body area normalized to the corresponding Vect group: values for PFF-fed groups are expressed as % of PFF-fed Vect control group, and values for Buffer-treated groups are expressed as % of Buffer-treated Vect control group. n = 28 for each group. Two-tailed *t* test. **(C)** Quantification of number of CEP cell bodies. Data are mean ± SEM. n = 7 for each group. One-way ANOVA with Dunnett's post hoc test. **(D)** Quantification of ADE cell body area normalized to the corresponding Vect group: values for PFF-fed groups are expressed as % of PFF-fed Vect control group, and values for Buffer-treated groups are expressed as % of Buffer-treated Vect control group. n = 14 for each group. Two-tailed *t* test. **(E)** Quantification of number of ADE cell bodies. Data are mean ± SEM. n = 7 for each group. One-way ANOVA with Dunnett's post hoc test. Vect, empty vector RNAi. *$P < 0.05$, **$P < 0.01$, ***$P < 0.001$. Boxplots show minimum, 25th percentile, median, 75th percentile, and maximum.

reported transfer of α-synuclein between pharyngeal muscle and neuronal cells, with disease-associated phenotypes (14). Cross-tissue transmission of α-synuclein has also been observed from dopaminergic neurons or muscle to the hypodermis (15). Although these studies demonstrate that α-synuclein can travel between cells and even tissues in *C. elegans*, in most cases it is not known if the transmission of α-synuclein is prion-like, that is, capable of seeding aggregation of α-synuclein in the recipient cell. Moreover, the nature of the α-synuclein species, that is monomers, oligomers, fibrils, or even physiological conformers, is often not known because of transgenic expression of α-synuclein in both the donor and recipient cells.

In contrast, our study uses biochemically well-defined aggregated species of α-synuclein as the initial seed, and demonstrates the induction of aggregation of host α-synuclein in a prion-like manner. In a recent study in which α-synuclein oligomers were fed to *C. elegans* and were able to cross the intestinal barrier and cause toxicity (28), the worms lacked host α-synuclein and thus these are not models of prion-like α-synuclein activity but rather provide insights into non-prion–like mechanisms of α-synuclein–initiated disease. In our models, we found that later time-points after PFF feeding, or exposure to high PFF concentrations, resulted in non-prion–like phenotypes in which monomeric α-synuclein treatment was similarly toxic to PFFs, and the phenotypes were not

progressive. It is possible that mechanisms engaged in these contexts might be similar to the non-prion–like events induced by exogenous oligomers in worms lacking α-synuclein (28).

In our study, we were able to define a set of conditions that produced prion-like disease phenotypes induced by feeding worms α-synuclein PFFs. In these paradigms, exposure to PFFs, but not monomer, caused dopamine neuron degeneration, and promoted the aggregation of host α-synuclein resulting in an associated progressive motor decline. For the strain expressing pan-neuronal α-synuclein, the optimal set of conditions were 0.5 ng/μl PFFs, and conducting assays on day 2. For the strain expressing α-synuclein in the muscle, the optimal set of conditions were 0.25 ng/μl PFFs, and conducting assays on days 2 through 4. The differences in these dosages and timings may reflect differences in α-synuclein expression levels, variations in the genetic backgrounds, and/or differences in the biology of the target tissues. In addition, although disease phenotypes in these cases require both PFFs and host α-synuclein, it is not yet known how PFFs reach target cell types in the body, for example, through simple diffusion in the pseudocoelomic fluid, or potentially more regulated cell-to-cell transmission mechanisms.

PFF feeding in *C. elegans* provides a new approach for the modeling of PD phenotypes and disease progression. The vast array of tools available for genetic manipulation in *C. elegans* and the

**A** α-synuclein PFFs

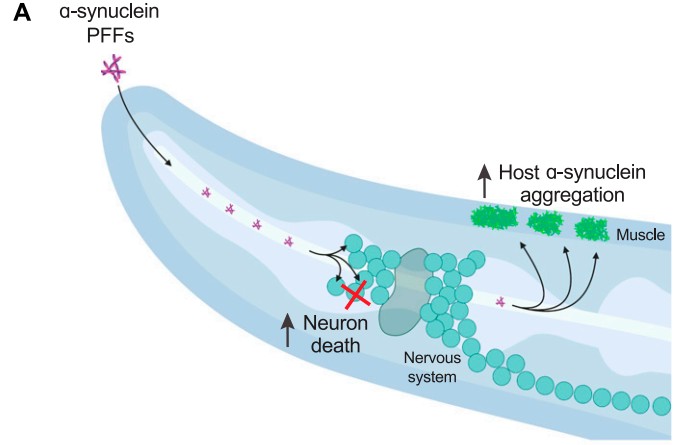

**B** α-synuclein PFFs

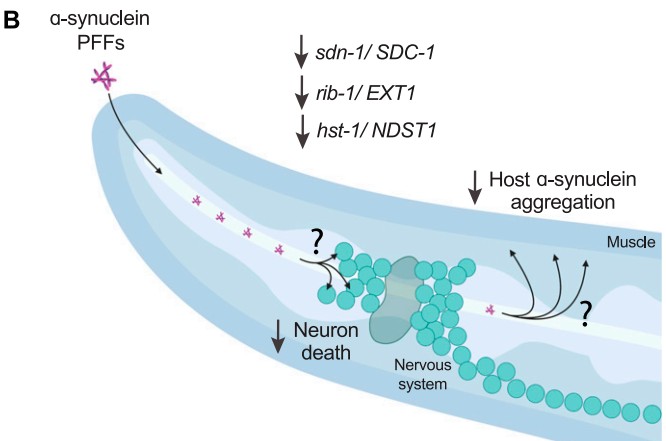

**Figure 7. Model figure.**
**(A)** Ingestion of α-synuclein pre-formed fibrils (PFFs) in *C. elegans* expressing human α-synuclein induces dopamine neuron cell death and seeds the aggregation of host α-synuclein. **(B)** RNAi-mediated knockdown of HSPG genes *sdn-1*, *rib-1*, or *hst-1*, rescues dopamine neurons from PFF toxicity and reduces PFF-driven aggregation of host α-synuclein. The mechanisms by which HSPG pathway components regulate PFF toxicity, i.e. potentially by facilitating PFF uptake or cell-to-cell transmission, remain unknown. Figure created with BioRender.com.

feasibility of large-scale investigations makes this an advantageous model system for the discovery of new disease mechanisms and potential therapeutic targets. We have identified specific components of the HSPG pathway as mediators of PFF neuronal and muscle cell injury in *C. elegans*. One of these factors, the cell surface proteoglycan, syndecan, has known roles in nervous system development and axon guidance (29), and was previously found to facilitate α-synuclein uptake into SH-SY5Y human neuroblastoma cells (30). It is possible that syndecan/*sdn-1* may serve as a PFF cell surface receptor in our in vivo models, especially given its ubiquitous expression in adult *C. elegans* tissues, including intestine, neurons, and muscle (31).

The specific involvement of some HSPG synthesis enzymes and not others in PFF disease phenotypes suggests that specific modification patterns may be required for PFF cellular uptake and/ or toxicity. We identified the glycosyltransferase *EXT1*/*rib-1* that is critical for the polymerization of heparan sulfate chains, and the

deacetylase and sulfotransferase *NDST1*/*hst-1*, as important regulators of PFF-driven phenotypes. Both genes are known to be expressed in adult *C. elegans* neurons and muscle (31). Both *EXT1* and *NDST1* were also previously reported to be required for α-synuclein internalization into HEK293T cells, and the binding of α-synuclein fibrils to heparins was found to depend on overall sulfation (32). *NDST1* acts upstream of other sulfotransferases and attaches *N*-sulfate groups on glucosamine residues. Loss of this early *N*-sulfation step likely affects downstream modifications and overall charge (32, 33), and may thereby afford protection in our PFF models potentially by disrupting HSPG structures that are necessary for α-synuclein toxicity and/or uptake into neurons or muscle. Together, these findings open up new avenues of investigation in a powerful model system for further discovery.

## Materials and Methods

### *C. elegans* strains

Worms were maintained at 20°C on standard nematode growth medium (NGM) plates or high growth medium (HG) plates, seeded with OP50 *E. coli* or HT115 RNAi *E. coli*, as indicated. The following strains were used in this study: wild-type N2 Bristol strain, NL5901 pkIs2386 [*unc-54p::hWTα-synuclein::yfp*], LC108 uIs69 [pCFJ90 *myo-2p::mCherry + unc-119p::sid-1*], UM0011 [*dat-1p::gfp; aex-3p::hWTα-synuclein*], AM134 rmIs126 [*unc-54p::yfp*], BY250 [*dat-1p:: gfp*], and MOR002 [*dat-1p::gfp; aex-3p::hWTα-synuclein*]; uIs69 [pCFJ90 *myo-2p::mCherry + unc-119p::sid-1*] obtained by crossing UM0011 with LC108.

### α-Synuclein PFF preparation and treatment

Recombinant human wild-type α-synuclein (Proteos) was incubated at 1 mg/ml at 37°C and shaking at 1,400 rpm using an Eppendorf Thermomixer R for up to 10 d. Fibrillization was assayed by Thioflavin T (Sigma-Aldrich) added to a final concentration of 25 μM. Fluorescence emission was measured at 482 nm during excitation at 450 nm. Sedimentation analysis was performed by centrifugation at 16,000*g* for 10 min at 4°C. The supernatants and pellets were boiled in SDS sample buffer at 95°C for 10 min, and run on SDS–PAGE gels (Invitrogen). The gels were then stained with Coomassie Blue R-250. When α-synuclein fibrillization had reached the plateau phase, and insoluble α-synuclein was present, PFFs were aliquoted and stored at −80°C until use. On the day of *C. elegans* treatment, PFFs were thawed at room temp, diluted to 0.1 mg/ml in PBS, and sonicated. Freshly sonicated PFFs were then mixed with OP50 at the indicated concentrations in a total volume of 300 μl M9, dispensed onto unseeded NGM plates and allowed to dry for 1–2 h before transferring to the plates day 1 adult worms that had been synchronized by bleaching. Monomer α-synuclein was prepared exactly in parallel, except with thawing on ice and without sonication. Buffer treatments used M9 in the absence of α-synuclein. The day after treatment, worms were transferred to new NGM plates that had been seeded with OP50 without α-synuclein. Thereafter, worms were transferred to new

OP50-seeded NGM plates without α-synuclein every 2 d for the duration of the experiment.

### Imaging Alexa Fluor–labeled PFFs

To image PFF seeds, freshly sonicated PFFs were labeled with Alexa Fluor 647 dye according to the protein labeling kit instructions (A30009; Thermo Fisher Scientific). As with unlabeled PFFs, 647-labeled PFFs were mixed with OP50, dispensed onto unseeded NGM plates, and the plates were allowed to dry for 1–2 h. Day 1 adult NL5901 worms were then transferred to 647-labeled PFF plates for 24 h. As indicated, worms were then given a 1 h "washout" period in which they were transferred to OP50-seeded NGM plates without PFFs to remove excess labeled PFFs from the digestive tract. Worms were mounted on 2% agarose pads with M9 and sodium azide, and imaged on a Nikon A1R MP+ multiphoton/confocal microscope at either 20× or 60× magnification to obtain z-stacks. For quantification of labeled PFFs in the lumen, intestinal tissue, non-digestive tissues in the head, and vulva, Cy5 signal intensity was measured in maximum intensity projections and the Cy5 signal of the background (obtained for each individual image from an area without worms) was subtracted. To quantify colocalization of labeled PFFs with α-synuclein–YFP in the muscle, a binary mask was created in NIH Elements software designating all YFP-positive pixels. The Cy5 signal was then quantified in all YFP-positive pixels and the Cy5 signal of the background was subtracted.

### Imaging α-synuclein–YFP aggregates

Worms expressing α-synuclein–YFP in muscle cells (strain NL5901) or, as a control, only YFP in muscle (strain AM134) were mounted on 2% agarose pads with M9 and sodium azide, and imaged on a Nikon A1R MP+ multiphoton/confocal microscope at 60× magnification to visualize host α-synuclein aggregation. Z-stacks were analyzed using Fiji software. Maximum intensity projections were generated, and an intensity threshold was set to distinguish aggregates (bright foci/puncta) from background. Once a threshold was set, it was used across all treatment groups in the experiment.

### Motor assays

At indicated ages, worms (strains NL5901 or N2) were individually picked onto unseeded NGM plates. Worms were allowed to recover for 1 min, and then the number of body bends was manually counted for 30 s. Worms were excluded if they did not move at all from their origin point.

### Neurodegeneration imaging

Worms expressing GFP in dopamine neurons (strains UM0011, MOR002, and BY250) were mounted on 2% agarose pads with M9 and sodium azide, and imaged on a Nikon A1R MP+ multiphoton/confocal microscope at 60× magnification. To quantify cell body area, z-stacks were analyzed using Fiji software. A region of interest was drawn around each cell body in the image slice corresponding to the cell's maximal size. All visible ADEs and CEPs were measured,

and an area size of zero was assigned to missing cells. A numerical threshold for area ($\mu m^2$) was set to distinguish cell body size that would be considered present/non-degenerated. Once a threshold was set, it was used across all treatment groups in the experiment. Neurites projecting from CEP cell bodies were scored using 3D rendering in NIS Elements software. Neurites were counted as either normal, blebbed, or fragmented. In all cases, LUTS settings were standardized across images.

### RNAi treatments

RNAi clones were obtained from the Ahringer RNAi library (Source BioScience, Cambridge University Technical Services Limited). Worms were synchronized by bleaching and plating onto HG plates seeded with OP50. At the L4 stage, worms were transferred to RNAi-seeded NGM plates containing carbenicillin and IPTG that were pre-induced with 0.1 M IPTG. The next day, freshly sonicated PFFs were mixed with RNAi bacteria at the indicated concentrations in a total volume of 300 $\mu$l M9, dispensed onto unseeded RNAi plates, induced with 0.1 M IPTG, and allowed to dry for 1–2 h before transferring the day 1 worms. Buffer treatments used M9 in the absence of α-synuclein. The day after treatment, worms were transferred to new RNAi plates that had been seeded without α-synuclein and were pre-induced with 0.1 M IPTG. Thereafter, worms were transferred to new RNAi plates without α-synuclein every 2 d for the duration of the experiment. Control vector (Vect) RNAi refers to empty vector L4440 in HT115 *E. coli*. L4440 was a gift from Andrew Fire (plasmid # 1654; Addgene; http://n2t.net/addgene:1654; RRID: Addgene_1654).

### Quantitative real-time PCR

NL5901 worms were fed HSPG RNAi or Vect control starting at L4. On day 1, worms were crushed in liquid nitrogen and added to Trizol LS reagent. RNA was isolated using chloroform-isopropanol extraction method, samples were DNase treated (QIAGEN), and cDNA was synthesized with an oligo dT primer and Superscript III First-Strand Synthesis System (Thermo Fisher Scientific). SYBR Green PCR Master Mix (Thermo Fisher Scientific) was mixed with cDNA and primers for *hst-2* (Fwd: ACGGTCCCCGACTTTTTCAA, Rev: AGTGGTATTGGAGCG-GAAGC) or *hse-5* (Fwd: ATGATGAAACAATGCGGGCG, Rev: TTCAGT-GATCGGACACCTGC), and *pmp-3* (Fwd: AGTTCCGGTTGGATTGGTCC, Rev: CCAGCACGATAGAAGGCGAT) was used as a reference gene. qPCR was performed on a Bio-Rad CFX96 Real-Time PCR Detection System, and gene expression was quantified using the ΔΔCt method.

### Statistical analysis

Statistical analyses were performed using GraphPad Prism 9. An unpaired two-tailed *t* test was used for all comparisons between two groups. For comparisons between multiple groups, one-way ANOVA or two-way ANOVA (for repeated measures or two-variable analyses, that is, motor assays with multiple treatment groups and multiple strains) was performed with post hoc testing as indicated. Experimenters were not blinded to the genotypes/groups when performing analyses.

# Supplementary Information

# Acknowledgements

We thank the *C. elegans* Genetics Center for strains (P40 OD010440), and the Department of Neuroscience and Regenerative Medicine 2-Photon Microscopy Facility. Strains N2 and NL5901 were generously provided by CT Murphy (Princeton University). Strain UM0011 was generously provided by G Wong (University of Macau) and C Wang (University of Hong Kong). Strain BY250 was generously provided by R Blakely (Florida Atlantic University). Strain AM134 was generously provided by R Morimoto (Northwestern University). This work was supported by the start-up fund from the Medical College of Georgia at Augusta University.

## Author Contributions

M Chen: conceptualization, formal analysis, investigation, and writing—original draft, review, and editing.
J Vincent: conceptualization, formal analysis, and investigation.
A Ezeanii: formal analysis and investigation.
S Wakade: formal analysis and investigation.
S Yerigenahally: formal analysis and investigation.
DE Mor: conceptualization, data curation, formal analysis, supervision, funding acquisition, investigation, visualization, methodology, project administration, and writing—original draft, review, and editing.

## Conflict of Interest Statement

The authors declare that they have no conflict of interest.

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
