## [Reviewer comments · Life Science Alliance]

Life Science Alliance

Heparan sulfate proteoglycans mediate prion-like α -synuclein toxicity in Parkinson's in vivo models

Merry Chen, Julie Vincent, Alexis Ezeanii, Saurabh Wakade, Shobha Yerigenahally, and Danielle Mor
DOI: <https://doi.org/10.26508/lsa.202201366>

Corresponding author(s): *Danielle Mor, Augusta University*

Review Timeline:

Submission Date:	2022-01-07
Editorial Decision:	2022-02-07
Revision Received:	2022-05-24
Editorial Decision:	2022-06-13
Revision Received:	2022-06-20
Accepted:	2022-06-22

Transaction Report:

February 7, 2022

Re: Life Science Alliance manuscript #LSA-2022-01366-T

Dr. Danielle E Mor
Augusta University
1120 15th Street
Augusta, Georgia 30912

Dear Dr. Mor,

Thank you for submitting your manuscript entitled "Heparan sulfate proteoglycans mediate toxic α -synuclein transmission in Parkinson's in vivo models" to Life Science Alliance. The manuscript was assessed by expert reviewers, whose comments are appended to this letter. We, thus, encourage you to submit a revised version of the manuscript back to LSA that responds to all of the reviewers' points.

Thank you for this interesting contribution to Life Science Alliance. We are looking forward to receiving your revised manuscript.

Sincerely,

B. MANUSCRIPT ORGANIZATION AND FORMATTING:

Reviewer #1 (Comments to the Authors (Required)):

Heparan sulfate proteoglycans mediate toxic α -synuclein transmission in Parkinson's in vivo models

In this manuscript, the authors are using a simple *C. elegans* model to test a potential interaction between "consumed" alpha-synuclein aggregates and internal phenotypes, which include aggregation in an orthogonal tissue, i.e., muscles, and degeneration of dopaminergic neurons, which are the vulnerable neuron types in humans with Parkinson's disease. Overall, the authors are trying to address some interesting questions and present some data to support their hypotheses. Specifically, they see an effect of feeding the preformed fibrils on the animals, and can genetically reverse those by targeting specific proteins in the HS synthesis pathway or a potential HSPG (syndecan). This is a basic characterization paper that presents a new and potentially powerful model for investigating gut-derived spread and propagation of α -synuclein aggregates. The effect of the fed particles was dependent on expression of alpha-synuclein in the animals, which was an important observation.

The data on the HSPG knockdown are convincing, independent of the concerns I have with the model, described below, there is something very interesting happening there. The authors have used a neural-specific RNAi approach, so that at least points to a site of action for follow up studies (not necessary here).

However, there is a fundamental disconnect in the model, at least with the evidence provided. Primarily, the suggestion of the prion-like seeding requires evidence that the PFFs can be documented to pass through the intestinal cells and into the body where they could encounter the transgenically expressed alpha-synuclein to induce the effects seen. There is an effect here, but the evidence for uptake is missing (see below), and therefore, the issues with this model need to be addressed, or at least potential alternative hypotheses presented. Below are some of the major concerns that should be considered.

Major Issues:

1) There are two major concerns I have about the feeding assay. First, based on the data presented in Figure 2, the PFFs fed stay in the lumen, and are not taken up by the intestinal cells. There is no convincing evidence for Alexa-stained particles in the body. The statement (P.6) "These findings suggest that PFFs spread from the digestive tract to the body tissues" is not supported by any evidence. More specifically, the lumen is where you would expect to find molecules not taken up by the intestinal cells. I would urge the authors to allow all of the dye to wash out of the lumen, for example, by buffer treatment for several hours) and then provide higher-zoom images of the intestinal cell bodies or muscles to provide any evidence that there is uptake.

In the absence of evidence that uptake does occur, the description of prion-like effects must be modified. There are a number of papers documenting the cell non-autonomous effects on neurons by inducing the unfolded protein response in the intestine. Given the evidence presented here, that explanation should at least be mentioned by the authors as an alternative hypothesis for their observations.

A potential alternate way to test this would be to use a different aggregate, that is not alpha-synuclein. That is, does a different aggregate (for example, the Q40YFP poly-glutamine polymer) induce the same effects? Such an observation would still be interesting, but would point to a non-direct interaction.

A second issue is the disparate effects of dosage and the presentation thereof. For example, the data in Figure 1 is presented with a single dose, and the lower dose is presented in a supplemental figure. However, in Figure 3, the data on dose-dependency are presented in a single figure. I am not sure why this is not done in Figure 1 as well. I would also like more information on how the initial dosage was determined, e.g., is there some idea of the relevant concentration from the other models or was it arbitrary?

The data in Figure 3 are also inconsistent with the title and description of results. The title of the figure "PFF feeding induces progressive motor decline" however, in panels B, C and D, the monomer is roughly equivalent to the PFF, and this needs to be addressed. Further, the only treatment that causes a "progressive" decline is 0.25 ng/ul, the others are not progressive, and that should also be stated more clearly.

In the figures where the authors are analyzing the cell bodies and neurites, the evidence for degeneration is not that strong. Counting the neurons is okay, although more evidence of membrane blebbing or cell death would be helpful. Could the authors discuss why they only see one ADE neuron in the controls? Is it a function of the stacking in z-projections?

Minor Issues:

1) Figure 1: I would suggest a cartoon indicating which neurons are the CEP and which are the ADEs to help non-experts make more sense of the anatomy.

- 2) Bar graphs are a poor way to document this phenotype. That is, it is impossible for an animal to have 0.8 neurons. I would encourage the authors to use individual data points with the number of each neuron observed (0-4 for CEPs, 0-2 for ADEs) to better present the spread of the data.
- 3) I think a depiction of the treatment would be helpful, as it is difficult to determine in each figure, when the PFF/MON treatment starts, when it concludes and when the assays are being performed.
- 4) It is important to be clear whether the experimenters were blind to the genotypes in doing their analyses.
- 5) As above with regard to an experimental "guide" the RNAi experiments should have an indication of when RNAi was initiated versus when the PFF/MON treatment was initiated. I think this will help the readers orient to the experiment.
- 6) The model shown in Figure 6 implies that the knockdown of HSPG genes is inhibiting uptake. Even if there were clear evidence of uptake, it is very difficult to understand how neuron-specific knockdown could affect the uptake and transcytosis of the PFFs. The effect on the aggregation observed by YFP in the muscles seems more solid.

Suggestions:

- 1) I think it would be best that the authors be a little more detailed in labeling figures. Figure 2 would be better presented if strain names were providing next to the images.
- 2) I think it would help the readers to have a bit more information about the differences in the muscle and neuronal expression lines, which might provide at least some guidance as to why the tissues are differentially susceptible to the doses. I can think of multiple potential issues, including the genetic backgrounds they were generated in, the expression levels of the alpha-synuclein in the different tissues, etc. I do not think the difference in tissue susceptibility is a concern, but some discussion would be helpful.

Reviewer #2 (Comments to the Authors (Required)):

In this paper, the authors show that exposure to alpha-synuclein pre-formed fibrils (PFF) leads to dopaminergic neurodegeneration and formation of alpha-synuclein aggregation in muscles in *C. elegans*. They also show that this requires internal alpha-synuclein, suggesting prion-like conversion of host alpha-synuclein. This work is interesting in that it establishes a convenient system to study alpha-synuclein seeding effect. Furthermore, the author proves the usefulness of this system by showing that RNAi knockdown of genes involved in heparan sulfate proteoglycan synthesis are involved in this process. However, there are some issues that need to be addressed, which I listed below.

Major points

1. Page 6 line 16. "These findings suggest that PFFs spread from the digestive tract to body tissues", Figure 6. The paper suggests that PFF first enters the digestive tract of *C. elegans*. But because animals were cultured in PFFs mixed with bacteria, other parts of animals, including cuticles, sensory neurons and the vulva, were exposed to PFFs and PFFs could enter from these cells. Authors showed that labelled PFFs exist in the intestinal lumen. But the lumen is outside of cells. Does labelled PFFs exist within intestinal cells or enteric neurons? If not, experiments to determine the involvement of digestive organs is needed for the conclusion, or the conclusions should be changed. Furthermore, in the dopaminergic neurodegeneration experiment, intestinal cells do not express alpha-synuclein, and in the muscle expression experiment, intestinal cells and enteric neurons do not express alpha-synuclein. If the spreading of misfolded alpha-synuclein require host alpha-synuclein, shouldn't intestinal cells be required to have alpha-synuclein? Or is the assumption here is that PFFs reach dopaminergic neurons and muscle without going into intestinal cells? Please clarify in the manuscript.
2. Title "Heparan sulfate proteoglycans mediate toxic α -synuclein transmission in Parkinson's in vivo models", Page 9 line 20. "These results suggest that *sdn-1*, *rib-1*, and *hst-1* play important roles in prion-like α -synuclein transmission from the digestive tract to neurons and muscle." It remains possible that these genes do not play roles in the transmission itself. These genes could be involved in misfolding of alpha-synuclein. If PFFs is indeed taken up from the intestine, these genes could be affecting intake of PFFs from outside of the body (pumping). Or maybe the structure of intestine is changed. There may be other possibilities. Authors should experimentally address if these genes are involved in transmission or weaken the conclusion.

Minor points

1. Page 12 line 9. "It is possible that syndecan/ *sdn-1* may serve as a PFF cell surface receptor in our in vivo models." Authors mention the possibility of SDN-1 working as a PFF receptor. Is *sdn-1* (and other HSPG pathway genes) expressed in the intestine, dopaminergic neurons, or muscle? If there are previous studies analyzing the expression pattern, please describe them in the text.
2. Page 13 line 2. Description of strains should follow the standard *C. elegans* nomenclature rules. Brackets, not parentheses, should be used to describe transgenes.
3. Page 15 line 15. "Maximum intensity projections were generated and an intensity threshold was set in order to distinguish aggregates from background." There is no explanation on how authors defined aggregates. Please describe the definition.
4. Page 21 figure legend for Fig 3. The concentration of PFF (0.25 ng/ul) for E, F, G should be mentioned.
5. Fig 4. In Fig 4A, the buffer control of RNAi-treated animals is should be shown. It is possible that these RNAi knockdown animals exhibit increased or decreased body bends in the absence of PFF. If these animals have altered baseline activity,

interpretation of the data need to be changed. Similarly, there is no buffer control in Fig 4C and D and Fig 5, either. Results of vector and RNAi animals in the absence of PFF should also be shown.

Reviewer #3 (Comments to the Authors (Required)):

In this manuscript, the authors describe that feeding wild-type human α -synuclein expressing *C. elegans* with α -synuclein pre-formed fibrils (PFFs) induced dopaminergic neurodegeneration and prion-like seeding of aggregation. The accelerated α -synuclein aggregation in *C. elegans* muscle cells was associated with a progressive motor deficit. RNAi-mediated knockdown of *sdn-1* and four heparan sulfate biosynthesis genes (*rib-1*, *rib-2*, *hst-1*, and *hst-3.2*), but not the other three heparan sulfate biosynthesis genes (*Hse-5*, *hst-2*, and *hst-3.1*), protected against PFF-induced α -synuclein aggregation and dopaminergic neurodegeneration. This work demonstrates human α -synuclein expressing *C. elegans* as a new *in vivo* model to investigate gut-derived α -synuclein spreading and propagation of Parkinson's disease, and heparan sulfate proteoglycans mediate the toxic α -synuclein transmission. The elucidation of heparan sulfate proteoglycan is a major molecule to mediate α -synuclein transmission is novel, but additional experiments are needed to strengthen the data and conclusions.

Major concern:

1. Fig. 2D. No α -synuclein aggregates are seen in the PFF-fed *C. elegans*, as shown in Fig. 2A. Pictures at higher magnification need to be included.
2. Fig. 4. The gene knockdown efficiency and heparan sulfate proteoglycan expression alteration must be determined (mRNA, protein, and heparan sulfate). These data are essential, especially the knockdown efficiency of the genes that did not show a protective effect. This paper will be significantly improved if the heparan sulfate structure data is available.

Minor concern:

1. Fig. 3B. On day 4, is the difference between buffer - and Mon-treatment groups significant?
2. Discussion: The statement "The specific involvement of some HSPG synthesis enzymes and not others in PFF disease phenotypes suggests that specific modification patterns may be required for PFF cellular uptake." This statement needs to be elaborated. The work reported by Stopschinski BE, et al. (Ref. 30) has shown that " τ aggregates display specific interactions with HSPGs that depend on GAG length and sulfate moiety position, whereas α -synuclein aggregates exhibit more flexible interactions with HSPGs." This work suggested that α -synuclein aggregate binding to heparan sulfate mainly depends on a negative charge. This appears to agree with the author's data. The authors might include how the gene knockdown alters heparan sulfate structure and overall charge in the discussion. This information can be found in a recent *Nat Method* paper from Qiu H., et al. (PMID: 30377379).

We wish to thank the reviewers for their extremely insightful comments. In light of this constructive feedback, we have heavily revised the manuscript and believe that it is significantly improved. Please see below for a point-by-point response to the reviewers' comments.

Reviewer #1:

In this manuscript, the authors are using a simple C. elegans model to test a potential interaction between "consumed" alpha-synuclein aggregates and internal phenotypes, which include aggregation in an orthogonal tissue, i.e., muscles, and degeneration of dopaminergic neurons, which are the vulnerable neuron types in humans with Parkinson's disease. Overall, the authors are trying to address some interesting questions and present some data to support their hypotheses. Specifically, they see an effect of feeding the preformed fibrils on the animals, and can genetically reverse those by targeting specific proteins in the HS synthesis pathway or a potential HSPG (syndecan). This is a basic characterization paper that presents a new and potentially powerful model for investigating gut-derived spread and propagation of alpha-synuclein aggregates. The effect of the fed particles was dependent on expression of alpha-synuclein in the animals, which was an important observation.

The data on the HSPG knockdown are convincing, independent of the concerns I have with the model, described below, there is something very interesting happening there. The authors have used a neural-specific RNAi approach, so that at least points to a site of action for follow up studies (not necessary here).

We thank the reviewer for their positive feedback regarding the utility of the new C. elegans models and the HSPG knockdown data.

However, there is a fundamental disconnect in the model, at least with the evidence provided. Primarily, the suggestion of the prion-like seeding requires evidence that the PFFs can be documented to pass through the intestinal cells and into the body where they could encounter the transgenically expressed alpha-synuclein to induce the effects seen. There is an effect here, but the evidence for uptake is missing (see below), and therefore, the issues with this model need to be addressed, or at least potential alternative hypotheses presented. Below are some of the major concerns that should be considered.

We completely agree; in the original submission, the evidence for spread of PFFs into body tissues was lacking. We have now added an entire figure (Fig 1) and supplemental figure (Fig S2) presenting data regarding PFF uptake, showing that PFFs are indeed detected in intestinal tissue and other body tissues. Please see below for more details.

Major Issues:

1) There are two major concerns I have about the feeding assay. First, based on the data presented in Figure 2, the PFFs fed stay in the lumen, and are not taken up by the intestinal cells. There is no convincing evidence for Alexa-stained particles in the body. The statement (P.6) "These findings suggest that PFFs spread from the digestive tract to the body tissues" is not supported by any evidence. More specifically, the lumen is where you would expect to find molecules not taken up by the intestinal cells. I would urge the authors to allow all of the dye to wash out of the lumen, for example, by buffer treatment for several hours) and then provide higher-zoom images of the intestinal cell bodies or muscles to provide any evidence that there is uptake.

We agree with the reviewer. Figure 1 of the manuscript is now focused on PFF uptake and quantification of Alexa-stained particles in body tissues. We have provided high-magnification images following a 1-

hour “washout” period, to remove excess PFFs from the digestive tract (1 hour is known to be sufficient for *C. elegans* to starve and clear the gut). We find that after this washout period, PFFs are detected in intestinal tissue, other body tissues, as well as colocalized with human α -synuclein in the muscle. These data are consistent with the spreading of exogenous PFFs into multiple body tissues.

In the absence of evidence that uptake does occur, the description of prion-like effects must be modified. There are a number of papers documenting the cell non-autonomous effects on neurons by inducing the unfolded protein response in the intestine. Given the evidence presented here, that explanation should at least be mentioned by the authors as an alternative hypothesis for their observations.

A potential alternate way to test this would be to use a different aggregate, that is not alpha-synuclein. That is, does a different aggregate (for example, the Q40YFP poly-glutamine polymer) induce the same effects? Such an observation would still be interesting, but would point to a non-direct interaction.

We thank the reviewer for these suggestions. We hope that the new data presented in Fig 1 and Fig S2 will satisfy the concerns.

A second issue is the disparate effects of dosage and the presentation thereof. For example, the data in Figure 1 is presented with a single dose, and the lower dose is presented in a supplemental figure. However, in Figure 3, the data on dose-dependency are presented in a single figure. I am not sure why this is not done in Figure 1 as well. I would also like more information on how the initial dosage was determined, e.g., is there some idea of the relevant concentration from the other models or was it arbitrary?

We apologize that the data on dosages was not consistent in the original submission. We have modified the organization and flow of all of the figures and hope that the rationale behind the dosages and the timing of testing in the different strains is now logical and clear to the reader. We also began this study using an arbitrary dosage of PFFs (there are no citations for PFF exposure in *C. elegans*) and titrated our experiments from there.

The data in Figure 3 are also inconsistent with the title and description of results. The title of the figure "PFF feeding induces progressive motor decline" however, in panels B, C and D, the monomer is roughly equivalent to the PFF, and this needs to be addressed. Further, the only treatment that causes a "progressive" decline is 0.25 ng/ul, the others are not progressive, and that should also be stated more clearly.

We apologize that the description of Figure 3 was not clear. We have modified the title of the figure and description of the results to reflect a more clear and accurate account of the data.

In the figures where the authors are analyzing the cell bodies and neurites, the evidence for degeneration is not that strong. Counting the neurons is okay, although more evidence of membrane blebbing or cell death would be helpful. Could the authors discuss why they only see one ADE neuron in the controls? Is it a function of the stacking in z-projections?

We agree that the neurodegeneration analysis was not sufficiently robust in the original manuscript. We have now re-analyzed all of the neuron images, and provided data for both cell body area (for both CEP and ADE neuron types), in addition to the number of each cell type. This more detailed analysis revealed PFF toxicity at an earlier time-point than was previously detected. We also provide more detailed analysis of dopaminergic neurites, including frequency of blebbing and fragmentation. Regarding ADE

neurons in controls, the worms expressing human α -synuclein have subtle neurodegeneration due to this genetic background. Wild-type worms typically have both ADEs intact, as our data reflects.

Minor Issues:

1) Figure 1: I would suggest a cartoon indicating which neurons are the CEP and which are the ADEs to help non-experts make more sense of the anatomy.

We have added labels indicating the CEP and ADE cell types, and hope that this will make the images clearer to readers.

2) Bar graphs are a poor way to document this phenotype. That is, it is impossible for an animal to have 0.8 neurons. I would encourage the authors to use individual data points with the number of each neuron observed (0-4 for CEPs, 0-2 for ADEs) to better present the spread of the data.

We agree. The graphs have been changed to show individual data points for the number of cells.

3) I think a depiction of the treatment would be helpful, as it is difficult to determine in each figure, when the PFF/MON treatment starts, when it concludes and when the assays are being performed.

We thank the reviewer for this feedback. We have made every effort in the main text and the figure legends to be as clear as possible regarding when the treatments occur and when the assays occur. In the new Figures 2 and 3, the timing of the treatment is given directly on the graphs to facilitate understanding of the experimental design. The other figures have become quite large, and therefore a visual depiction of the experiments was not added to those due to lack of space.

4) It is important to be clear whether the experimenters were blind to the genotypes in doing their analyses.

The experiments were not conducted blindly, and a statement of this has been added to the methods section.

5) As above with regard to an experimental "guide" the RNAi experiments should have an indication of when RNAi was initiated versus when the PFF/MON treatment was initiated. I think this will help the readers orient to the experiment.

Due to lack of space in the figures, we could not add a schematic depicting the experimental design. However, we have made every effort to make the experimental paradigm as clear as possible in the main text, the figure legends, and the methods section.

6) The model shown in Figure 6 implies that the knockdown of HSPG genes is inhibiting uptake. Even if there were clear evidence of uptake, it is very difficult to understand how neuron-specific knockdown could affect the uptake and transcytosis of the PFFs. The effect on the aggregation observed by YFP in the muscles seems more solid.

We agree with the reviewer that it is important to remain agnostic as to where exactly the HSPG components are acting to promote PFF toxicity. The model therefore shows HSPG gene knockdown in general (with gene names positioned above the worm, in order to avoid any implication of specific localization), and we have added question marks to signify that the role in transmission is unclear.

Suggestions:

1) I think it would be best that the authors be a little more detailed in labeling figures. Figure 2 would be better presented if strain names were providing next to the images.

We have added additional labels of PFF dosage and timing, in addition to adding strain information into the figure legends.

2) I think it would help the readers to have a bit more information about the differences in the muscle and neuronal expression lines, which might provide at least some guidance as to why the tissues are differentially susceptible to the doses. I can think of multiple potential issues, including the genetic backgrounds they were generated in, the expression levels of the alpha-synuclein in the different tissues, etc. I do not think the difference in tissue susceptibility is a concern, but some discussion would be helpful.

We agree greater discussion of these points is necessary. These issues are now addressed in the discussion section.

Reviewer #2:

In this paper, the authors show that exposure to alpha-synuclein pre-formed fibrils (PFF) leads to dopaminergic neurodegeneration and formation of alpha-synuclein aggregation in muscles in C. elegans. They also show that this requires internal alpha-synuclein, suggesting prion-like conversion of host alpha-synuclein. This work is interesting in that it establishes a convenient system to study alpha-synuclein seeding effect. Furthermore, the author proves the usefulness of this system by showing that RNAi knockdown of genes involved in heparan sulfate proteoglycan synthesis are involved in this process. However, there are some issues that need to be addressed, which I listed below.

We thank the reviewer for their positive feedback regarding the new *C. elegans* models.

Major points

1. Page 6 line 16. "These findings suggest that PFFs spread from the digestive tract to body tissues", Figure 6. The paper suggests that PFF first enters the digestive tract of C. elegans. But because animals were cultured in PFFs mixed with bacteria, other parts of animals, including cuticles, sensory neurons and the vulva, were exposed to PFFs and PFFs could enter from these cells. Authors showed that labelled PFFs exist in the intestinal lumen. But the lumen is outside of cells. Does labelled PFFs exist within intestinal cells or enteric neurons? If not, experiments to determine the involvement of digestive organs is needed for the conclusion, or the conclusions should be changed.

We thank the reviewer for this feedback and completely agree these issues needed to be addressed. We have now added an entire figure (Figure 1) and supplementary figure (Fig S2) showing that PFFs are detected in intestinal tissue, other body tissues, and even found to be colocalized with human α -synuclein in the body wall muscle. While it is possible that PFFs are entering through non-digestive routes, as the reviewer suggested, we found that Alexa-labeled PFFs are not detected in the vulva tissue. It is also unlikely that the PFFs enter through the cuticle due to known impermeability of the adult *C. elegans* cuticle (a reference regarding this has been added to the main text).

Furthermore, in the dopaminergic neurodegeneration experiment, intestinal cells do not express alpha-synuclein, and in the muscle expression experiment, intestinal cells and enteric neurons do not express alpha-synuclein. If the spreading of misfolded alpha-synuclein require host alpha-synuclein, shouldn't intestinal cells be required to have alpha-synuclein? Or is the assumption here is that PFFs reach dopaminergic neurons and muscle without going into intestinal cells? Please clarify in the manuscript.

The reviewer's point is well taken, and we apologize that the manuscript was not clear regarding the role of host α -synuclein. We have observed that the toxicity of PFFs (PFF-induced disease phenotypes) require host α -synuclein expression in neurons or muscle, but it is not known how the PFFs reach these target tissues. We have now included a statement in the discussion regarding this point. We have also been extremely careful throughout the manuscript not to make assumptions or overstate the data with regards to α -synuclein transmission in these models.

*2. Title "Heparan sulfate proteoglycans mediate toxic α -synuclein transmission in Parkinson's in vivo models", Page 9 line 20. "These results suggest that *sdn-1*, *rib-1*, and *hst-1* play important roles in prion-like α -synuclein transmission from the digestive tract to neurons and muscle." It remains possible that these genes do not play roles in the transmission itself. These genes could be involved in misfolding of alpha-synuclein. If PFFs is indeed taken up from the intestine, these genes could be affecting intake of PFFs from outside of the body (pumping). Or maybe the structure of intestine is changed. There may be other possibilities. Authors should experimentally address if these genes are involved in transmission or weaken the conclusion.*

We completely agree with the reviewer. Being that the exact role of HSPG genes in PFF toxicity phenotypes is not known, we have changed the title to remain more agnostic. The new title reads: "Heparan sulfate proteoglycans mediate prion-like α -synuclein toxicity in Parkinson's *in vivo* models".

Minor points

*1. Page 12 line 9. "It is possible that syndecan/*sdn-1* may serve as a PFF cell surface receptor in our in vivo models." Authors mention the possibility of *SDN-1* working as a PFF receptor. Is *sdn-1* (and other HSPG pathway genes) expressed in the intestine, dopaminergic neurons, or muscle? If there are previous studies analyzing the expression pattern, please describe them in the text.*

A description of known tissue-specific expression for *sdn-1* and the other HSPG genes has been added to the discussion.

*2. Page 13 line 2. Description of strains should follow the standard *C. elegans* nomenclature rules. Brackets, not parentheses, should be used to describe transgenes.*

This has been corrected.

3. Page 15 line 15. "Maximum intensity projections were generated and an intensity threshold was set in order to distinguish aggregates from background." There is no explanation on how authors defined aggregates. Please describe the definition.

A description of how aggregates are defined has been added to the methods.

4. Page 21 figure legend for Fig 3. The concentration of PFF (0.25 ng/ul) for E, F, G should be mentioned.

This has been corrected.

5. Fig 4. In Fig 4A, the buffer control of RNAi-treated animals should be shown. It is possible that these RNAi knockdown animals exhibit increased or decreased body bends in the absence of PFF. If these animals have altered baseline activity, interpretation of the data need to be changed. Similarly, there is no buffer control in Fig 4C and D and Fig 5, either. Results of vector and RNAi animals in the absence of PFF should also be shown.

We completely agree. RNAi knockdown experiments in the absence of PFFs have now been added for all the figures mentioned (now found in Figures 5, 6, and supplementary figures 5 and 6).

Reviewer #3:

*In this manuscript, the authors describe that feeding wild-type human α -synuclein expressing *C. elegans* with α -synuclein pre-formed fibrils (PFFs) induced dopaminergic neurodegeneration and prion-like seeding of aggregation. The accelerated α -synuclein aggregation in *C. elegans* muscle cells was associated with a progressive motor deficit. RNAi-mediated knockdown of *sdn-1* and four heparan sulfate biosynthesis genes (*rib-1*, *rib-2*, *hst-1*, and *hst-3.2*), but not the other three heparan sulfate biosynthesis genes (*Hse-5*, *hst-2*, and *hst-3.1*), protected against PFF-induced α -synuclein aggregation and dopaminergic neurodegeneration. This work demonstrates human α -synuclein expressing *C. elegans* as a new *in vivo* model to investigate gut-derived α -synuclein spreading and propagation of Parkinson's disease, and heparan sulfate proteoglycans mediate the toxic α -synuclein transmission. The elucidation of heparan sulfate proteoglycan is a major molecule to mediate α -synuclein transmission is novel, but additional experiments are needed to strengthen the data and conclusions.*

We thank the reviewer for their positive feedback.

Major concern:

*1. Fig. 2D. No α -synuclein aggregates are seen in the PFF-fed *C. elegans*, as shown in Fig. 2A. Pictures at higher magnification need to be included.*

We have now included higher magnification images and made an entire figure (Figure 1) and supplementary figure (Fig S2) with a focus on PFF imaging. We have also added a time-course of α -synuclein aggregation with dose-response and high magnification images in what is now Figure 2.

2. Fig. 4. The gene knockdown efficiency and heparan sulfate proteoglycan expression alteration must be determined (mRNA, protein, and heparan sulfate). These data are essential, especially the knockdown efficiency of the genes that did not show a protective effect. This paper will be significantly improved if the heparan sulfate structure data is available.

We agree that gene knockdown efficiency is important to determine, particularly for those genes that did not produce a phenotypic effect. To this end, we have added quantitative real-time PCR data and confirmed that *hse-5* and *hst-2* genes (two out of the three genes that did not have a phenotypic effect) were successfully knocked down. For the third gene, *hst-3.1*, we could not confirm successful

knockdown by qPCR, and antibodies against this *C. elegans* protein do not exist. Therefore, we have removed the data for this gene from the manuscript.

Minor concern:

1. Fig. 3B. On day 4, is the difference between buffer - and Mon-treatment groups significant?

There was no statistically significant difference between these groups, and we have now explicitly noted this on the revised graph in Figure 3.

2. Discussion: *The statement "The specific involvement of some HSPG synthesis enzymes and not others in PFF disease phenotypes suggests that specific modification patterns may be required for PFF cellular uptake." This statement needs to be elaborated. The work reported by Stopschinski BE, et al. (Ref. 30) has shown that "tau aggregates display specific interactions with HSPGs that depend on GAG length and sulfate moiety position, whereas α -synuclein aggregates exhibit more flexible interactions with HSPGs." This work suggested that α -synuclein aggregate binding to heparan sulfate mainly depends on a negative charge. This appears to agree with the author's data. The authors might include how the gene knockdown alters heparan sulfate structure and overall charge in the discussion. This information can be found in a recent Nat Method paper from Qiu H., et al. (PMID: 30377379).*

We thank the reviewer for this insightful comment, and have now expanded the discussion on HSPGs to include how the gene knockdown may affect structure.

June 13, 2022

RE: Life Science Alliance Manuscript #LSA-2022-01366-TR

Dr. Danielle E Mor
Augusta University
Department of Neuroscience & Regenerative Medicine
1120 15th Street
Augusta, Georgia 30912

Dear Dr. Mor,

Thank you for submitting your revised manuscript entitled "Heparan sulfate proteoglycans mediate prion-like α -synuclein toxicity in Parkinson's in vivo models". We would be happy to publish your paper in Life Science Alliance pending final revisions necessary to meet our formatting guidelines.

- please add the Twitter handle of your host institute/organization as well as your own or/and one of the authors in our system
- please use the [10 author names, et al.] format in your references (i.e. limit the author names to the first 10)
- please add a callout for Figure S3A, D to your main manuscript text
- please consider uploading Figure 7 as a Graphical Abstract as well

A. FINAL FILES:

B. MANUSCRIPT ORGANIZATION AND FORMATTING:

Sincerely,

Reviewer #1 (Comments to the Authors (Required)):

The authors have substantially improved the quality of the data and presentation. They have satisfactorily responded to my previous critiques. The observed model is interesting, and the genetic interaction with HSPGs makes this a substantial move forward in the field.

Reviewer #2 (Comments to the Authors (Required)):

I have no further comments.

Reviewer #3 (Comments to the Authors (Required)):

In this revision, the authors addressed all my concerns by adding new data, reorganizing the presenting data and elaborating the discussion section. I do not have any further concerns about the paper.

June 22, 2022

RE: Life Science Alliance Manuscript #LSA-2022-01366-TRR

Dr. Danielle E Mor
Augusta University
Department of Neuroscience & Regenerative Medicine
1120 15th Street
Augusta, Georgia 30912

Dear Dr. Mor,

Thank you for submitting your Research Article entitled "Heparan sulfate proteoglycans mediate prion-like α -synuclein toxicity in Parkinson's in vivo models". It is a pleasure to let you know that your manuscript is now accepted for publication in Life Science Alliance. Congratulations on this interesting work.

DISTRIBUTION OF MATERIALS:

Again, congratulations on a very nice paper. I hope you found the review process to be constructive and are pleased with how the manuscript was handled editorially. We look forward to future exciting submissions from your lab.

Sincerely,
